# Comparative Cognition Research Demonstrates the Similarity between Humans and Other Animals

**DOI:** 10.3390/ani13071165

**Published:** 2023-03-25

**Authors:** Thomas R. Zentall

**Affiliations:** Department of Psychology, University of Kentucky, Lexington, KY 40506, USA; zentall@uky.edu

**Keywords:** comparative cognition, sameness, equivalence, transitive inference, justification of effort, gambling

## Abstract

**Simple Summary:**

The traditional view of cognition is that it is something that differentiates humans from other species. There is growing evidence, however, that if one “asks” other animals appropriately, they often show evidence that they too are capable of cognitive processes thought to be unique to humans. Furthermore, several behaviors thought to involve complex human cognition that are presumed to be culturally learned can be shown to occur in other species as well. Results such as these suggest that simpler processes are likely to be involved. Together, these lines of research demonstrate that the cognitive distinctions that have been made between humans and other animals may be greatly exaggerated. The present review focuses on several cognitive phenomena that can be demonstrated in animals. The review focuses primarily on pigeons (because they are not thought to be especially cognitive compared to primates) out of convenience, because they are a highly visual species. A better understanding of the similarities between humans and other animals should help foster improved treatment of nonhuman species, as well as, in some cases, improved treatment of disordered human behavior.

**Abstract:**

The field of comparative cognition represents the interface between the cognitive behavior of humans and other animals. In some cases, research demonstrates that other animals are capable of showing similar cognitive processes. In other cases, when animals show behavior thought to be culturally determined in humans, it suggests that simpler processes may be involved. This review examines research primarily with pigeons (out of convenience because of their visual ability). I start with the concept of sameness and follow with the concept of stimulus equivalence, the building blocks of human language. This is followed by research on directed forgetting, the cognitive ability to maintain or forget information. A hallmark of cognition is transitive inference performance (if A < B, and B < C, the understanding that A < C), but the variety of species that show this ability suggests that there may be simpler accounts of this behavior. Similarly, experiments that demonstrate a form of cognitive dissonance in animals suggest that dissonance may not be necessary to explain this biased behavior. Furthermore, examples of sunk cost in pigeons suggests that the human need to continue working on a failing project may also have a biological basis. Finally, pigeons show a preference for a suboptimal choice that is similar to unskilled human gambling, a finding that may clarify why humans are so prone to engage in this typically losing activity.

## 1. Introduction

The well-being of animals should concern us, if for no other reason than they are sentient beings. In addition, however, although there are clearly important differences in cognitive ability, there is evidence that they are more like us than we humans may be willing to acknowledge. In this review, based largely on research with pigeons (for convenience because of their visual acuity), I attempt to demonstrate some of that similarity by showing that certain cognitive processes thought to be unique to humans are also present in nonhuman species. Furthermore, I propose that some behaviors that have been taken as evidence that humans are different from other animals may be attributable to simpler processes that can also be found in other species. These examples are purposively quite varied to show the diversity of cognitive, and thought to be cognitive, behaviors that have parallels in humans and other animals.

Animal research has a rich history in the psychology of learning. Major contributions to our understanding of learning were made by researchers who focused on learning by animals (e.g., [1,2,3]). However, the “cognitive revolution” resulted in a separation between mainstream research involving the learning of humans and other animals. Although Skinnerian approaches to learning [2] are still used in many venues—behavior analysis is widely practiced in institutional, educational, and correctional institutions—cognitive psychologists have tended to focus on the more complex processes that humans are capable of demonstrating.

Nonetheless, the field of comparative cognition has developed in parallel with research on human cognition. Borrowing and adapting procedures from cognitive and developmental psychologists, animal researchers have discovered that many species are capable of much more than that which traditional behaviorists have generally given them credit. The attraction by comparative psychologists to research conducted by developmental psychologists came about because development psychologists are often interested in early learning by children, before they have fully developed human language. Because developmental psychologists cannot count on their subjects’ understanding instructions or producing appropriate verbal responses indicating that the instructions are understood, researchers have had to rely on integrating instructions as part of the task. Unlike cognitive psychologists, who could give human adults explicit task instructions (e.g., recall as many items as you can from the first list you learned), the challenge for both developmental and comparative psychologists is to incorporate the task instructions into the task itself. Unfortunately, when one finds that subjects perform poorly on these tasks, these researchers cannot be sure if the failure represents the absence of the ability being studied or the failure to “understand” the task instructions.

Developmental psychologists have created a number of ingenious procedures that have provided insights into children’s early understanding of their environment. For example, when young nonverbal children are shown an impossible or unexpected event, they tend to look at it longer (a procedure known as the violation of expectation, e.g., [4]), and it has been shown that dogs do as well [5]. Thus, the challenge for comparative cognitive psychologists is to explore the cognitive behavior of nonverbal animals using appropriate experimental designs. In this vein, Macphail has proposed a radical theory [6] that much of the assessed differences among vertebrate animals can be attributed to (1) differences in the sensory abilities of different species, (2) differences in the way that their responses are measured, or (3) differences in their motivation to engage in the task presented to them. Whether this proposition is correct or not, it correctly identifies three important components of any task that must be arranged to maximize the chances that the animal will be able correctly perform the task presented.

Our anthropocentric perspective often gets in the way of accurately assessing the abilities of many species. A classic example comes from Harlow’s [7] research on learning sets or learning to learn. When monkeys are given a set of simultaneous three-dimensional object discriminations, one at a time, they generally improve their learning as the number of discriminations increase. With sufficient experience, they can show excellent learning on the second trial of a new discrimination. Rats, however, show only minimal improvement over a similar set of object discriminations. However, care must be taken to maximize the conditions under which the animal might understand the task instructions. Rats, being essentially nocturnal animals, do not have the visual acuity of primates, and they rely largely on their other senses. When a similar experiment was conducted using odors as the discriminative stimuli, Slotnick and Katz [8] found improvement over discriminations that rivaled that of the monkeys. Thus, rats too showed clear evidence of learning to learn.

In this review, I have selected several cognitive behaviors that I have studied and are presumed to be primarily human. The aim of the research is to ask if other animals also show evidence of these cognitive behaviors. In some cases, the research suggests that other animals are capable of similar cognitive behavior. In other cases, simpler mechanisms likely underlie what appears to be complex cognitive behavior. Importantly, simpler mechanisms may underlie what has been taken to be cognitive behavior in humans.

## 2. The Sameness Concept

Do pigeons recognize the sameness relation between two stimuli? Skinner [9] famously proposed that they do not. He proposed that in a matching-to-sample task in which pigeons are presented with a sample stimulus and they have to choose the comparison stimulus that matches the sample, the pigeons learn a stimulus response chain involving the sample, the correct comparison response, and the reinforcer. However, many researchers have shown that, under the right conditions, pigeons show high levels of same/difference discrimination. For example, Katz and Wright [10] presented pigeons first with one stimulus (a color photograph) to which they were required to peck, and then another stimulus. If the two stimuli matched, a response to the second one was reinforced. If the two stimuli did not match, a response to a white square to the side of the stimulus pair was reinforced. Not only do pigeons show excellent learning of this task with a large number of colored photographic slides, but they also show excellent transfer to new slides.

Another approach to the question of sameness learning by pigeons was used by Wasserman et al. [11], who presented pigeons with a matrix of iconic stimuli. If all the icons matched, a response to one side location was reinforced. If all the icons were different, a response to the other side location was reinforced. The pigeons readily learned this task and transferred their learning to novel stimuli. Furthermore, when the number of matching and mismatching stimuli in the matrix were varied, the pigeons showed a graded, psychophysical response to the mixture of same and different icons.

Using a simpler design involving fewer stimuli, Zentall and Hogan [12] trained pigeons on a two-shape (circle and plus) matching or mismatching task (for mismatching, the pigeons were rewarded for selecting the comparison stimulus that did not match the sample). When the pigeons had learned their respective task, all of them were transferred to red and green stimuli involving the same relation for half of the pigeons, but the other relation for the remaining pigeons (see Figure 1 for the design of this experiment). Zentall and Hogan found significant savings for the pigeons that were transferred to the same relation as in training compared with the pigeons that were transferred to the other relation (see Figure 2).

Recently we have found a challenge to another of Skinner’s [9] proposals that the sameness relation between stimuli does not play a role for pigeons [13]. In this study, pigeons were trained on a two-color conditional discrimination in which the center (sample) stimulus indicated which of two side (comparison) stimuli was correct. Red samples were presented with red and yellow comparison stimuli, yellow samples were presented with yellow and green comparison stimuli, green samples were presented with green and blue comparison stimuli, and blue samples were presented with blue and red comparison stimuli (see Figure 3). Some of the pigeons were trained to match the sample (e.g., if the sample was red, choose the red comparison stimulus), whereas other pigeons were trained to choose the mismatching comparison stimulus (e.g., if the sample was red, choose the yellow comparison stimulus). Thus, for both groups, during training each color had served as the sample as the correct comparison stimulus and as the incorrect comparison stimulus. On test trials, a stimulus replacement procedure was used in which either the matching or the mismatching comparison stimulus was replaced by one of the colors not experienced with that sample. The results indicated that pigeons that were trained to match the sample had learned to select the matching comparison, whereas pigeons that were trained to mismatch the sample had learned to reject the matching comparison stimulus (Figure 4; [13]).

Thus, under the right conditions, pigeons can demonstrate that sameness is a quite natural concept, and they use the concept to select or reject the stimulus that matches the sample. That pigeons have a natural bias to match is not surprising, given that if they encounter a promising food item they would likely be “on the lookout” for other similar food items.

## 3. Acquired Equivalence

The phenomenon of acquired equivalence is a form of symbolic association in which, for example, for humans, when an object and a word mean the same thing, whatever one learns about either, one likely will transfer to the other. For example, a young child may learn from experience that dogs are friendly, playful animals, yet one hopes that telling the child that not all dogs are friendly and some dogs may bite may make the child cautious in the presence of novel dogs. In this case, the relation between the word and the object should form an equivalence set, and if it does, it may allow what the child learns about dogs from the words of the adult to affect the way that the child reacts to a novel dog.

Under laboratory conditions, we have found that such equivalence relations can be shown to occur in pigeons with rather arbitrary stimuli [14]. For example, if in a conditional discrimination, on different trials, either of two conditional stimuli can serve as a signal to respond to the same comparison stimulus, an equivalence relation appears to develop between the conditional stimuli (see Figure 5, top). The equivalence relation can be shown when the pigeons learn a new association between one of the conditional stimuli and a new comparison stimulus (Figure 5, middle). One often finds that the new relation transfers to the second conditional stimulus without further training (Figure 5, bottom). In the example given, in the presence of a red light or a vertical line, a response to a large circle but not a small circle is reinforced, but in the presence of a green light or a horizontal line, a response to a small circle but not a large circle is reinforced. Now, if the red light then comes to signal that a response to a blue light (but not a white light) is reinforced and the green light signals that a response to the white light (but not the blue light) is reinforced, one finds that the pigeons will tend to choose the blue light in the presence of the vertical line and choose the white light in the presence of the horizontal line. That is, the response that was trained to the red and green lights has transferred to the vertical and horizontal lines, respectively, without further training. This result suggests that the red stimulus and vertical line stimulus have formed an equivalence set and the green stimulus and the horizontal line stimulus have also formed an equivalence set. The stimuli in an equivalence set have the property of having a similar meaning to each other. Equivalence relations form the basis of much human language learning (words have a meaning similar to the objects that they represent). For example, one can describe a property of an object using the word for that object and it will likely transfer to the object itself. Thus, equivalence research with pigeons suggests that the building blocks of a complex language (if not the complex language itself) are present in other animals as well.

## 4. Directed Forgetting

Cognitive research can be characterized as the search for evidence of active, controlled processes. Although consensus on what constitutes a cognitive process is not easily found, a cognition may be viewed as involving an active process that intervenes between the stimulus and response to modify that relation according to the task demands [15].

One cognitive process that has been studied in humans is directed forgetting [16,17]. If humans are told that they will not have to remember certain items that are presented, it reduces their ability to remember them at a later time (compared with items that they are instructed to remember). Directed forgetting has been studied in humans under a variety of procedures, but all of them involve the presentation of written or verbal material followed by signals to forget selected items and remember others. The typical finding is that items that are signaled to be forgotten are not remembered as well as items that are signaled to be remembered. This procedure generally involves telling subjects that they can forget certain items, but then testing them on those items anyway. It is assumed that humans actively process (rehearse) items that they were instructed to remember, but not items that they were instructed to forget, and that active rehearsal process makes those items more available to be recalled.

Directed forgetting has been studied in nonhuman animals using a version of a conditional discrimination. In a conditional discrimination, a sample stimulus signals which of two comparison stimuli is correct [18,19]. The challenge in studying directed forgetting in nonverbal animals is to find a way to incorporate the instructions into the task. Many of the experiments have started by training pigeons on a delayed conditional discrimination in which the sample stimulus indicates which of two comparison stimuli is correct. For example, if the sample is red, it indicates that choice of the red comparison will be reinforced, and if the sample is green, it indicates that choice of the green comparison will be reinforced. To study the animal’s ability to remember the sample, a delay has been inserted between the offset of the sample and the onset of the comparison stimuli [20,21]. Typically, research has shown that memory for the sample decreases monotonically as the delay increases [15].

A problem in studying directed forgetting in nonhuman animals is how to present them with instructions to either “remember” or to “forget.” Furthermore, the instructions have to be trained as a part of the task. This has been done with pigeons by inserting stimuli during the delay interval. For example, if vertical lines appear during the delay following the sample, it indicates that at the end of the delay, the comparison stimuli will appear, and the choice of the correct comparison stimulus will result in reinforcement. Thus, the vertical lines can be thought of as a “remember” cue because they serve as a signal that the pigeon will have to remember the color of the sample. If horizontal lines appear on the sample key during the delay, however, it indicates that the comparison stimuli will not appear at the end of the delay. Thus, the horizontal lines can be thought of as a “forget” cue because they serve as a signal that the pigeon will not have to remember the color of the sample. Directed forgetting is then tested on probe trials on which a forget cue is followed by a memory test (see Figure 6). The finding that performance on such probe trials is poorer than on trials on which a remember cue is presented is generally interpreted as evidence for directed forgetting [22].

When pigeons have been tested on probe trials with comparison stimuli following the forget cue, a substantial deficit in accuracy has typically been found [19,23,24,25,26]. There is a problem with this procedure, however, in interpreting such effects as a simple loss of memory. Although the forget cue signals the absence of a test for sample memory, it also signals the end of the trial. That is, it signals that a reinforcer cannot be obtained. Thus, the remember and forget cues also have differential motivation value, and the deficit in conditional discrimination accuracy associated with the forget cue that has been attributed to loss of memory for the sample may also result from the low motivational value associated with the forget cue (it signals that they will not get fed).

The differences in motivational value of the remember and forget cues can be largely remedied by using what has come to be called a substitution procedure, in which on forget cue trials, some event substitutes for the comparison stimuli presented on remember cue trials. Perhaps the best-substituted event has been a simple simultaneous discrimination involving stimuli different from the conditional discrimination and the remember and forget cues [23,27,28]. On these substitution trials, following a forget cue, one of the comparison stimuli is always correct, independent of the sample (see Figure 7). That is, following the forget cue, the pigeons still have to choose between the two stimuli for the stimulus that is correct, but memory for the sample is not required. The substituted events following a forget cue in training control for motivational and other non-memorial effects. Interestingly, under these conditions, directed forgetting has not typically been found. Thus, when differences in the motivational value of the remember and forget cues are controlled for, there appears to be no evidence for directed forgetting in pigeons.

Roper, Kaiser, and Zentall [29] suggested, however, that the reason the pigeons did not show a directed forgetting effect when motivational differences and other artifacts were controlled for was human subjects are often given items in the form of a list, with cues after each item designating whether the item is to be remembered or forgotten. Because the list procedure involves both remember and forget items within the list, when the subjects encountered a forget cue, they could use the time before the next item was presented to rehearse prior items that they were instructed to remember. Thus, for humans, the forget cue allows subjects to reallocate their active processing to remember items.

Could it be that for pigeons, on a given trial with a single item to remember, the forget cue had little effect on memory for the sample because there was no remember item to which to redirect memory? Roper et al. [29] designed a task, depicted in Figure 8, in which there were two possible forget cues (e.g., a large or a small circle). In this design, the forget cue instructed the pigeon to forget the sample color and remember the size of the circle, because when the circle was no longer present, vertical and horizontal lines were presented as comparison stimuli, and to obtain reinforcement, the pigeon had to remember which circle had been presented. Now, on forget cue probe trials, after presentation of either a large or a small circle, instead of line comparison stimuli, red and green comparison stimuli were presented, and to obtain a reinforcer, the pigeon was required to choose the comparison stimulus that matched the sample color that was presented at the start of the trial. Under these conditions, in spite of having the appropriate controls for motivation and other non-memorial factors, pigeons showed a significant deficit in forget cue probe trials as compared with remember cue trials. Thus, under these conditions, pigeons do show evidence for a directed forgetting effect. Evidence for directed forgetting suggests that memory for pigeons may not be an automatic process but may be under at least some cognitive control.

## 5. Transitive Inference

The ability to form a transitive inference implies that if one knows the relation between two objects, A and B, as well as the relation between B and a third object C, one should be able to infer the relation between A and C [30]. Thus, for a child to form a transitive inference if she is given the following propositions: Alice is taller than Betty, and Betty is taller than Carol, and she is then asked, “Who is taller, Alice or Carol?” she should be able to make the inference that Alice is taller.

Piaget assumed that children would have to have reached the concrete operational stage of development (about 7 years old) before they would understand the rules of logic and be able to solve this problem. Bryan and Trabasso [31] proposed, however, that the difficulty of this problem for younger children may not have been because of deficits in logic, but rather from deficits in memory for the propositions. Furthermore, if the relation being studied was, for example, height, with A the tallest, C the shortest, and B in between, the logical moderator B would not be needed to solve the problem, because A is always tall, and C is never tall. To deal with both of those problems, Bryan and Trabasso gave extensive training on the premise pairs using rods of different lengths and used four premise pairs—AB, BC, CD, and DE—rather than two (AB and BC). With four premise pairs, they could avoid testing with either endpoint, A (always tall) or E (never tall), but they could still test with the untrained pair B and D (both sometimes tall and sometimes short). With this procedure, Bryan and Trabasso found that children as young as 4 years old showed evidence of transitive inference.

Using the Bryan and Trabasso design [31], McGonigle and Chalmers [32] reasoned that one could test transitive inference in nonverbal animals by training them on a series of simultaneous discriminations of the kind A+ B-, B+ C-, C+ D-, D+ E-, in which plus meant choice of that stimulus was reinforced, whereas minus meant choice of that stimulus would not be reinforced. Nondifferentially reinforced test trials with B versus D would test for transitive inference, because those stimuli had appeared as both reinforced and nonreinforced stimuli and they had not appeared together during training. Using this procedure, McGonigle and Chalmers found evidence for transitive inference not only in young children but also in squirrel monkeys.

Since then, transitive inference has been found in many other species, including chimpanzees [33], rats [34], pigeons [35], pinyon jays [36], hooded crows [37], baby chicks [38], and even fish [39]. The widespread finding of transitive inference in many animal species raised the possibility that the ability to form transitive relations may have evolved to facilitate the learning of social hierarchies [40]. For example, if Animal X has learned that it is subordinate to Animal B and it has learned from observation that Animal B is subordinate to Animal A, it would be useful for Animal X to infer that it is also subordinate to Animal A. Although that explanation of the transitive inference effect found in the laboratory using arbitrary stimuli (e.g., colored lights) makes for a good story, it has always seemed like a remarkable generalization from social hierarchy in nature in which the target animal has a position in the series, to arbitrary stimuli in the laboratory in which the target animal is not one of those stimuli. Furthermore, if this social hierarchy account is correct, one would expect to see the transitive inference effect only in highly social animals that would find such a predisposed ability important, and it has been readily found in pigeons, a species not known to be especially social. Various models of transitive inference have been proposed, and the bulk of the evidence appears to favor a relatively simple associative strength model (see next section).

## 6. Cognitive Models of Transitive Inference

Two cognitive models of transitive inference have been proposed. One is based on the successive application of logical “if, then” rules [41]. On test trials involving the novel BD pair, subjects would go through the premise pairs in order until they reach the logical solution. Alternatively, Terrace [42] and D’Amato [43] suggested that during training, animals may integrate the premise pairs into an ordered series of mental (perhaps linear spatial) representations. On test trials involving the novel BD pair, it is assumed that the animal would consult the ordered representation and report which one is closer to the requested endpoint.

If the series is expanded to six or seven orderable stimuli, these two models make different predictions concerning the distribution of errors as a function of the tested pair of stimuli. If the subject must go through the premise pairs to arrive at a transitive solution, the farther apart the test pair is in the order, the more premise pairs would have to be consulted and the more likely an error would be made. If, however, during training the premise pairs are ordered into a spatial representation, the farther apart the test pairs are, the easier it should be to detect which one is closer to the endpoint in question.

In general, it has been found that as the number of intervening stimuli between the two test stimuli increases, accuracy also increases. This phenomenon, known as the symbolic distance effect, thus favors the spatial representation model.

## 7. Noncognitive Models of Transitive Inference

Given the large number of species that show the transitive inference effect, several investigators have suggested that a noncognitive mechanism might be involved. Several noncognitive theories have been proposed, including value transfer theory [35] and several even simpler reinforcement-based models [40,44].

## 8. Value Transfer Theory

According to value transfer theory, in a simultaneous discrimination, in addition to the direct value that a stimulus acquires from reinforcement when it is chosen, some of the value that accrues to the positive (reinforced) stimulus will transfer to the negative (nonreinforced) stimulus with which it was paired. One can think of this effect as resulting from a kind of spatial generalization (stimuli that are physically close to stimuli that have been associated with reinforcement will, to some extent, also be associated with reinforcement). According to value transfer theory, the amount of value that transfers will be proportional to the value of the positive stimulus with which the nonreinforced stimulus is paired. Thus, in the series A+ B-, B+ C-, C+ D-, D+ E-, during training, the B stimulus has been paired with the always positive A stimulus, whereas the D stimulus has been paired with the C stimulus, which was sometimes positive, in the C+ D- training pair, but sometimes negative, in the B+ C- training pair. Thus, in the A+ B-, B+ C-, C+ D-, D+ E- series, according to value transfer theory, the A stimulus has more value to transfer to the B stimulus than the C stimulus has to transfer to the D stimulus. Hence, on BD test trials, the B stimulus should be preferred over the D stimulus.

Evidence in support of value transfer theory was found by Zentall and Sherburne [45]. They trained pigeons on a simple simultaneous discrimination represented by A + B- and a second simultaneous discrimination represented by C± D- (in which choice of the C stimulus was reinforced 50% of the time). They then gave pigeons a choice between the B and D stimuli, neither of which had had received direct reinforcement. In this experiment, consistent with value transfer theory, on BD test trials, Zentall and Sherburne found that the pigeons preferred the B stimulus over the D stimulus.

However, value transfer theory has some problems accounting for the results of other research. For example, in an attempt to equalize value transfer in a transitive inference experiment, Weaver et al. [46] trained pigeons on an A± B-, B+ C±, C± D-, D+ E± series. For this group of pigeons, the direct value of A should be comparable to the direct value of C. Thus, the value that transfers from A to B should be comparable to the value that transfers from C to D. For this group of pigeons, although symmetrical value transfer should have occurred to B and D, stimulus B was still preferred over stimulus D. That is, transitive inference was still found.

## 9. Reinforcement-Based Theories

In a thorough review of the transitive inference literature, Vasconcelos [47] describes several reinforcement-based models. Each of the models is based on the history of reinforcement (and its absence) associated with each of the test stimuli. The simplest model is based on the Rescorla–Wagner [44] learning model. Wynne [40] refined the Rescorla–Wagner model by adding a configural component that gave the training pairs additional weight because they were presented together. With appropriate selection of the right parameters, these reinforcement-based models do a reasonable job of fitting most of the published data. Thus, it appears that one need not attribute the results of a transitive inference effect found in animals to a cognitive process.

But what about humans? When adult humans are tested on the transitive inference task, it is assumed that they use some form of logical cognitive inference [48]. However, it turns out that a logical inference may not be involved. For example, Green et al. [49] trained a group of humans who were not informed about the nature of the task. They were informed that they should try to learn the task by trial and error (a procedure that mimicked the procedure used with nonhuman animals). They found that when given the BD test pair, the human subjects, just like the other animal subjects, showed a reliable choice of the B stimulus. However, surprisingly, their transitive performance did not correlate with an awareness of the hierarchical relationship among the stimuli (as indicated by a postexperiment questionnaire). That is, the subjects did not appear to be aware of the logical order of the stimuli. The results of this experiment suggest that even in humans, transitive inference choices may not be based on logical inferences. Instead, inferences that appear to be based on logic inference may actually be based on simpler associative mechanisms about which human subjects are not aware.

## 10. Cognitive Biases

A cognitive bias in humans is thought to be a systematic thought process caused by the tendency of the human brain to simplify information processing through a filter of personal experience and preferences, rather than an objectively rational decision. Cognitive dissonance is a cognitive bias often defined as the discrepancy between one’s beliefs and one’s behavior [50]. It is typically attributed to the desire to be seen as consistent in one’s beliefs and one’s behavior (i.e., the desire not to be hypocritical).

If such biases can be found in other animals, it would suggest that these biases may not depend on complex human social experiences. Furthermore, if such biases can be found in other animals, it suggests that if one is interested in reducing or eliminating the effect of these biases on human decision-making processes, a different approach may be needed.

In general, it would be difficult to study the beliefs of a nonverbal animal; however, justification of effort is a form of cognitive dissonance found in humans that does not require a verbal response. Justification of effort occurs when there is a preference for an outcome that requires more effort to obtain over similar outcomes that require less effort [51,52]. Social psychologists have proposed that this preference results from the rationalization of the dissonance between one’s belief in the law of least effort and the behavioral effort that one has expended [49]. If justification of effort is based on cognitive dissonance, one would not expect to find it present in other animals. We have found, however, that under a variety of conditions, pigeons show a very similar effect [53], for example, if on some trials pigeons had to work hard (make 20 pecks) to receive a signal for food but on other trials less work (only one peck) was required to obtain a different signal for the same food (see Figure 9). Under these conditions, if pigeons are given a choice between the two signals, one would think that there should be no preference between the two signals because they both signal the same outcome. Should there be a preference, one might predict that the signal that occurs in training on trials in which less effort was required might be preferred because those trials represent “easy” effort. Instead, the pigeons prefer the signal that in training they had to work harder to obtain. The source of this preference is not clear, but Clement et al. [53] suggested that the justification-of-effort effect may result from a mechanism simpler than cognitive dissonance (i.e., simpler than, “If I had to work harder for the signal for reinforcement, the reward that followed must have been worth more”).

Clement et al. [53] suggested that the mechanism responsible for the preference for the stimulus that the pigeon had to work harder to obtain might be positive contrast. If the signal that comes after hard work represents positive contrast (in human terms it might represent relief or a reduction in frustration after hard work), those signals are likely to be preferred over signals for reinforcement that do not involve such positive contrast.

If positive contrast is responsible for the preference for the signal that follows hard work, it suggests that other relatively aversive events that precede signals for reward could have similar effects. We have found support for this prediction. For example, if a delay is inserted between the initial response and the signal for reinforcement, pigeons show a preference for that signal over another signal that was not preceded by a delay [54]. Similarly, on some trials, pecks to the start signal were followed by a reinforcer and then a signal for a second reinforcer, but on other trials, pecks to the start signal were followed by the absence of a reinforcer and then a different signal for a reinforcer. Following training, when the pigeons were given a choice between the two signals, they showed a preference for the signal for reinforcement that was followed by the absence of a reinforcer [55]. Thus, the release from a relatively aversive event appears to give greater value to the signal for reinforcement that follows.

Although the procedures used with pigeons may seem removed from justification-of-effort effects that have been found with humans, there is evidence that when humans are trained with procedures very similar to those of the pigeons, both children [56] and adult humans [57] show very similar effects. Furthermore, after choosing on test trials, when human adults were asked why they chose the stimulus leading to reward that they did, they rarely mentioned that it was the stimulus that was the one that they had to work harder to obtain during training [57]. That is, human adults are often unaware of the relation between the added effort and their choice of stimulus. Thus, it appears that the mechanism underlying choice between two stimuli that should have equal value (because they both result in the same reward) is likely implicit and is learned without awareness. Even when subjects seem to be aware of the relation between their effort and the stimulus that follows, they appear to be unaware that the effort that preceded the signal was the reason for their preference for the stimulus that followed greater effort [57].

The results of research with pigeons on the justification of effort effect (as well as other cognitive biases (see e.g., base-rate fallacy in pigeons; [58])) suggest that cognitive biases may have simple biological bases, and in humans, cultural factors unrelated to the reward itself may only strengthen these already existing predispositions.

## 11. Sunk Cost

A sunk cost is an expenditure of resources that has already occurred and cannot be recovered. Logically, one should not allow prior losses to affect current decisions. Instead, one should evaluate the objective likelihood of future outcomes without necessarily considering past outcomes. When one allows a sunk cost to determine the future investment of resources, it is referred to as the sunk cost fallacy. For example, humans often continue to invest in a failing business because to give up would be to lose the investment they have already made in the business, but objectively, that investment has already been lost, and further investment will likely lead to further losses.

This phenomenon would appear to be based on the human cultural value to avoid wasting resources, yet it can be shown that pigeons show a similar effect [59]. In that research, pigeons were first trained that the appearance of a green light signals that reinforcement requires 30 pecks, but the appearance of a red light signals that only 10 pecks are required for reinforcement. On test trials, after making a variable number of pecks to the green light—say, 10 pecks—the pigeons were given the option of completing the remaining 20 pecks to the green light or switching to the red light, which would require only 10 pecks for reinforcement. Although the optimal decision would be to switch to the red light, pigeons generally stayed with the green light, a decision that required more pecks and an increased delay of reinforcement. Thus, similarly to humans, pigeons appear to give added value to task completion. This result, too, suggests that cultural factors may reinforce existing tendencies, but they are not responsible for them.

## 12. Unskilled Gambling

When humans are engaged in unskilled gambling (e.g., slot machines, lotteries, roulette) their choice is almost always suboptimal (i.e., their investment is almost always greater than the return on their investment). People who engage in that form of gambling claim that they do so because it is entertaining. They often say that it is not for the money, but for the fun.

Optimal foraging theory suggests that animals have evolved to forage for food in the most efficient way possible. That is, animals that forage efficiently should live longer and reproduce more often. Thus, if given a choice between an infrequent but high-value reward and an optimal, more frequent, but lower-value reward, an animal should learn to choose the optimal outcome.

We have found, however, that pigeons and rats behave very much like problem gamblers. They show an attraction to the suboptimal low-frequency high-value “jackpot” over the optimal, high-frequency, lower-value outcome [60]. In that experiment, pigeons could choose the suboptimal alternative, for which there was a 20% chance of obtaining a green light that was always followed by 10 pellets of food, but an 80% chance of obtaining a red light that was followed by no pellets of food. Alternatively, they could choose the optimal alternative, for which there was a 100% chance of obtaining a yellow (or blue light) that was always followed by three pellets of food (see Figure 10). By choosing the suboptimal alternative, the pigeons indicated that they preferred an average of two pellets of food over a sure three pellets of food.

Later research suggested that for the pigeons, when they chose the low-frequency, high-value alternative and lost, it did not have the effect of a loss [61]. That is, the signal for a loss did not become a conditioned inhibitor. In human terms, it is as if there were an expectation of losing, so losing does not have the negative value that it should have. Furthermore, the value of the win appears to be greater than the reward itself [62]. That is, given a choice between a 50% chance of obtaining a signal for reinforcement (or obtaining a signal for the absence of reinforcement) and a 100% chance of obtaining a signal for reinforcement, pigeons actually showed a significant preference for 50% reinforcement over 100% reinforcement. It may be that the excitement of the win is what unskilled gamblers mean by the entertainment value of gambling, and for pigeons, it appears to have a similar effect.

Interestingly, with regard to the sunk cost effect described earlier, there is a phenomenon that human gamblers call chasing losses [63]. Chasing losses refers to the fact that gamblers often expend more money trying to make up for losses already incurred, rather than recognizing that they will likely only lose more money.

We have also examined some of the demographics of problem gamblers. For example, problem gamblers are often those who can least afford to lose money. What about pigeons? Our research suggests that although hungry pigeons tend to choose suboptimally, less hungry pigeons tend to choose optimally—they are less hungry, but they get more food [64].

Another characteristic of problem gamblers is they generally do not engage in other behavior that might give them comparable “enjoyment.” Gambling appears to be their sole source of pleasure. As it turns out, although laboratory pigeons often live in small cages, one bird to a cage, with limited opportunity for physical or social enrichment, we have found that giving pigeons a few hours a day in a large cage with other pigeons significantly reduces their tendency to choose suboptimally [65]. This finding may have implications for the treatment of human problem gambling. If it is the excitement of winning or even the thought of winning that motivates problem gamblers, it may be possible to provide problem gamblers with other exciting activities (e.g., canoeing, rock climbing, or even walks in nature) that might substitute for the pleasures associated with gambling.

## 13. Conclusions

The research described here offers but a few examples of the kind of cognitive research conducted with animals. Comparative cognitive research has advanced well beyond what we are typically exposed to in an undergraduate or graduate learning course. The breadth of comparative cognitive research lends credence to Macphail’s [6] brash proposition that the cognitive behavior of animals is largely an untapped resource. More importantly, however, the similarity of biases found in humans and other animals suggests that the underlying source of those biases may be based on basic evolved mechanisms that affect the behavior of many organisms, rather than factors unique to human evolution or the influence of culture. If so, research with animals may suggest alternative ways of dealing with related human problems.

## Figures and Tables

**Figure 1 animals-13-01165-f001:**
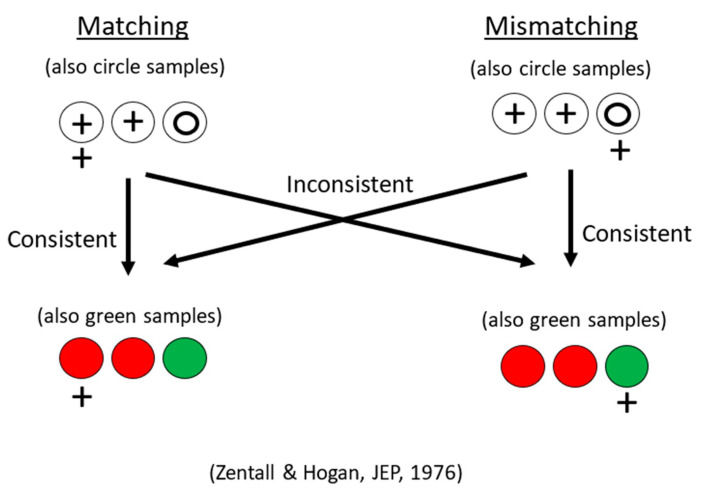
Design of same/different transfer experiment. Pigeons were trained on circle-plus matching or mismatching. They were all transferred to red and green stimuli, half to the same concept they were trained on (matching to matching or mismatching to mismatching), half to the other concept (matching to mismatching or mismatching to matching). After Zentall and Hogan (1976) [12].

**Figure 2 animals-13-01165-f002:**
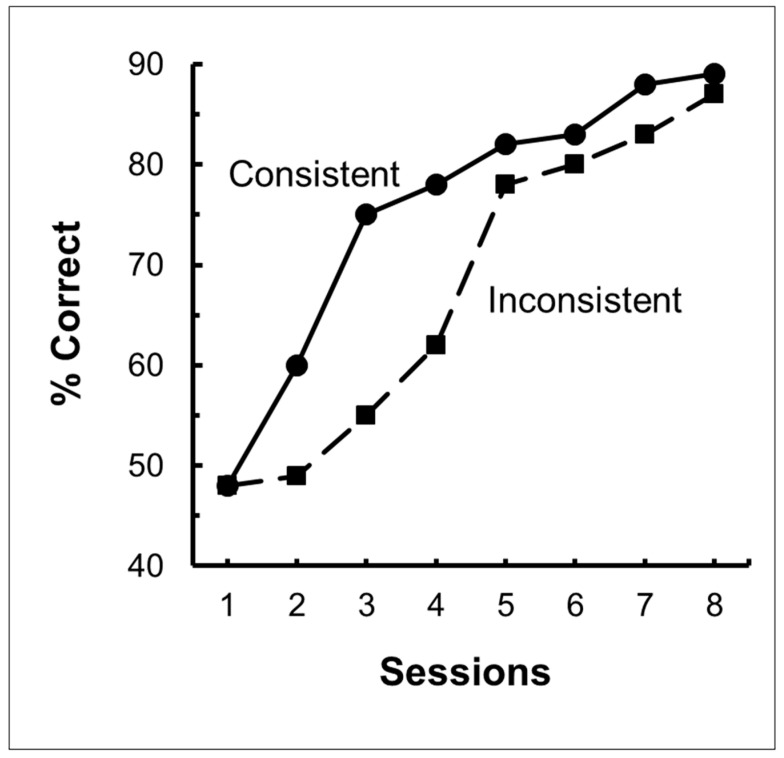
Results of same/different transfer experiment. Pigeons were trained on circle-plus matching or mismatching. They were all transferred to red and green stimuli, half to the same concept they were trained on (consistent), half to the other concept (inconsistent). After Zentall and Hogan (1976) [12].

**Figure 3 animals-13-01165-f003:**
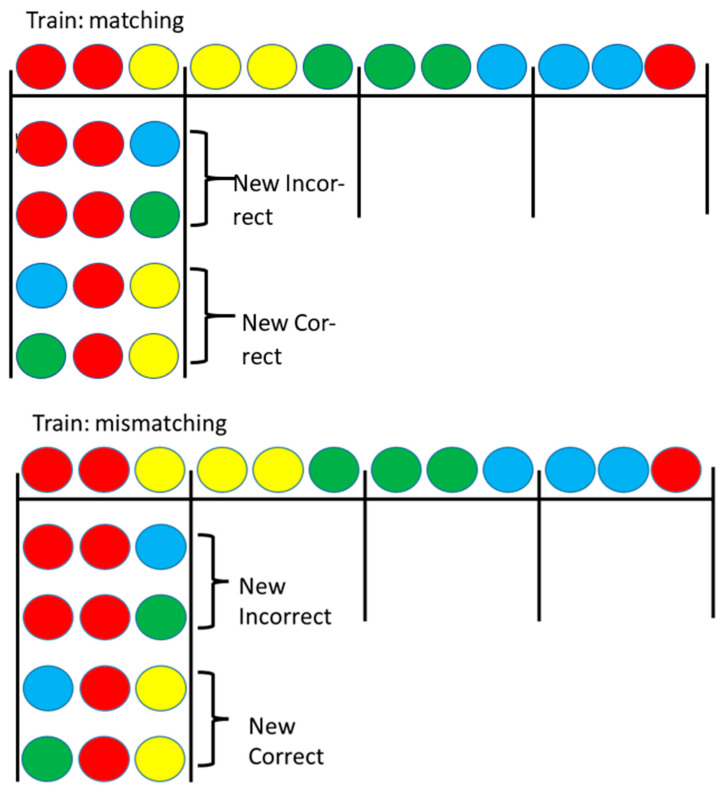
Pigeons were trained on four-color matching (**top**) or mismatching (**bottom**), as shown. Then, either the incorrect stimulus was replaced with a stimulus not seen with that sample during training or the correct stimulus was replaced with a stimulus not seen with the sample during training. Test trials for the yellow, green, and blue samples not shown. After Zentall et al. (2018) [13].

**Figure 4 animals-13-01165-f004:**
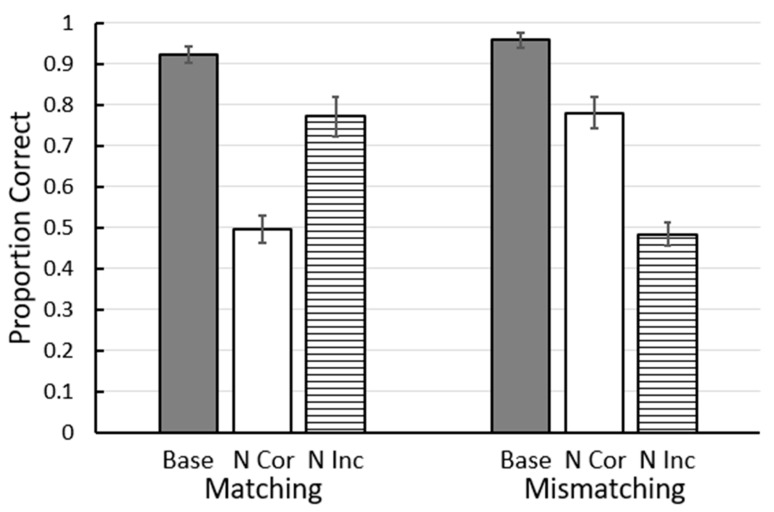
Results of Zentall et al. (2018) [13]. Base = baseline accuracy on matching or mismatching trials. N Cor = accuracy when the correct stimulus was replaced with a new stimulus. N Inc = accuracy when the correct stimulus was replaced with a new stimulus. Error bars = ± 1 standard error of the mean.

**Figure 5 animals-13-01165-f005:**
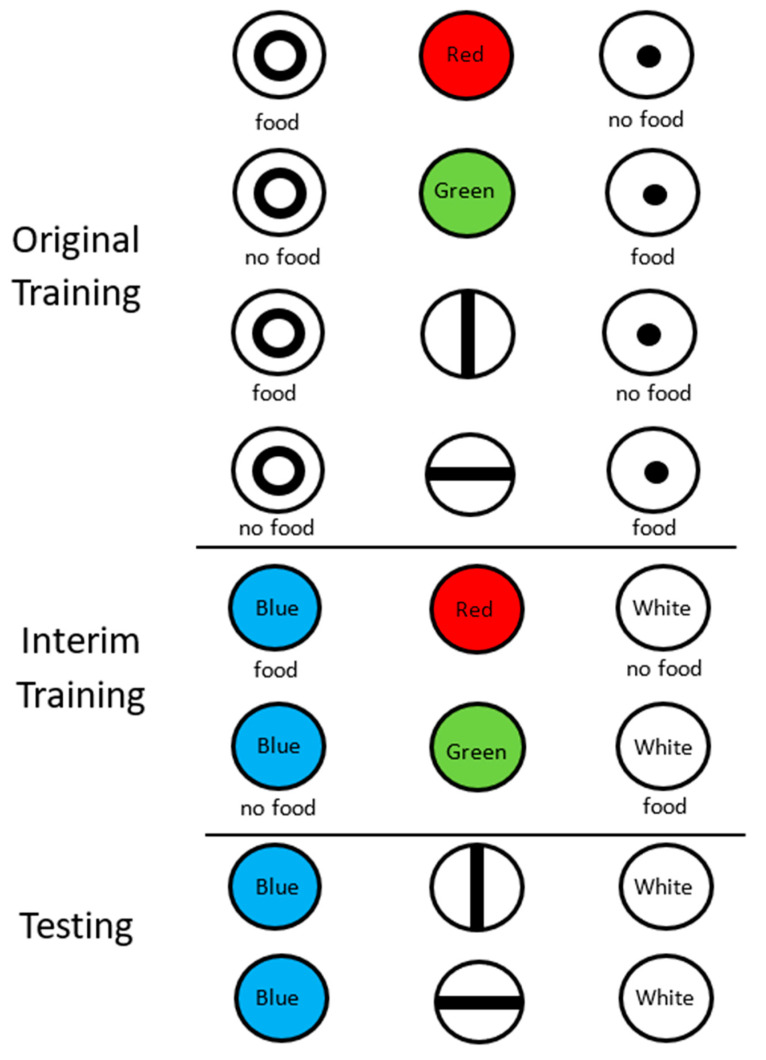
Design of the stimulus equivalence experiment (Urcuioli et al., 1989 [14]). In original training, pigeons were trained to select the large circle when the sample was red or a vertical line and to select the small circle when the sample was green or a horizontal line. During interim training, they were trained to select blue when the sample was red and white when the sample was green. On test trials, they were shown vertical and horizontal lines and they selected between blue and white.

**Figure 6 animals-13-01165-f006:**
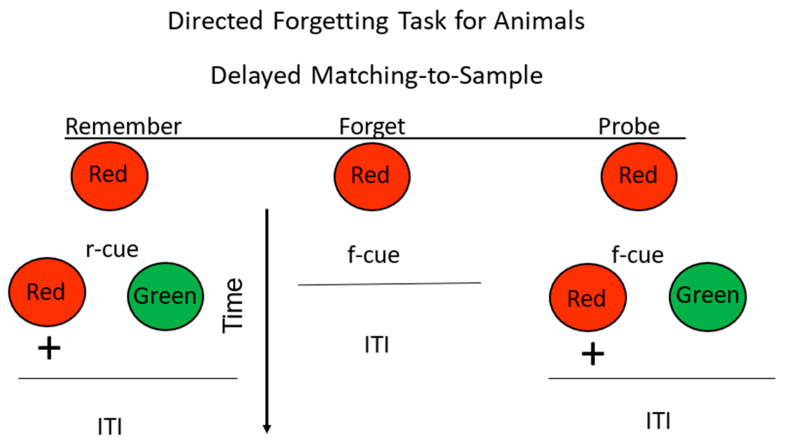
Directed forgetting: omission procedure. Pigeons were trained on a red/green delayed matching task. + = reinforcement. During the delay, insertion of a remember cue (r-cue) indicated that they would be tested for their memory for the sample. Insertion of a forget cue (f-cue) indicated that they would not be tested for their memory for the sample. On probe trials, they were presented with an f-cue, but were tested for their memory for the sample (after Maki and Hegvik, 1980 [23], omission group).

**Figure 7 animals-13-01165-f007:**
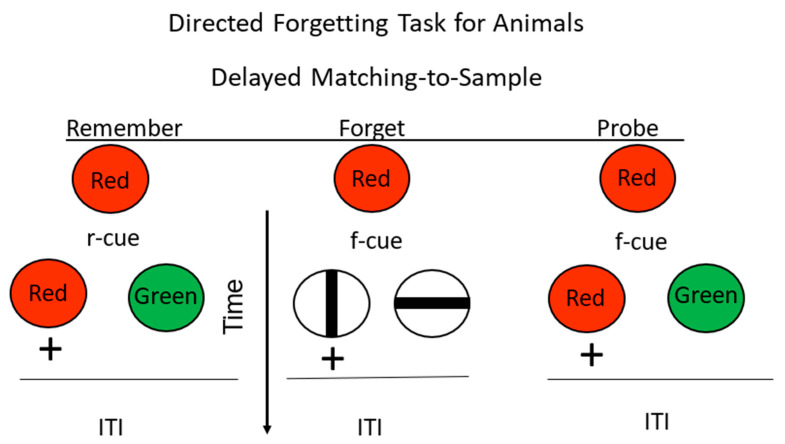
Directed forgetting: substitution procedure. Pigeons were trained on a red/green delayed matching task. + = reinforcement. During the delay, insertion of an r-cue (remember cue) indicated that they would be tested for their memory for the sample. Insertion of an f-cue (forget cue) indicated that they would not be tested for their memory for the sample, but they would have a simultaneous discrimination with reinforcement for selection of the vertical line. On probe trials, they were presented with an f-cue, but were tested for their memory for the sample (after Maki and Hegvik, [23] substitution group).

**Figure 8 animals-13-01165-f008:**
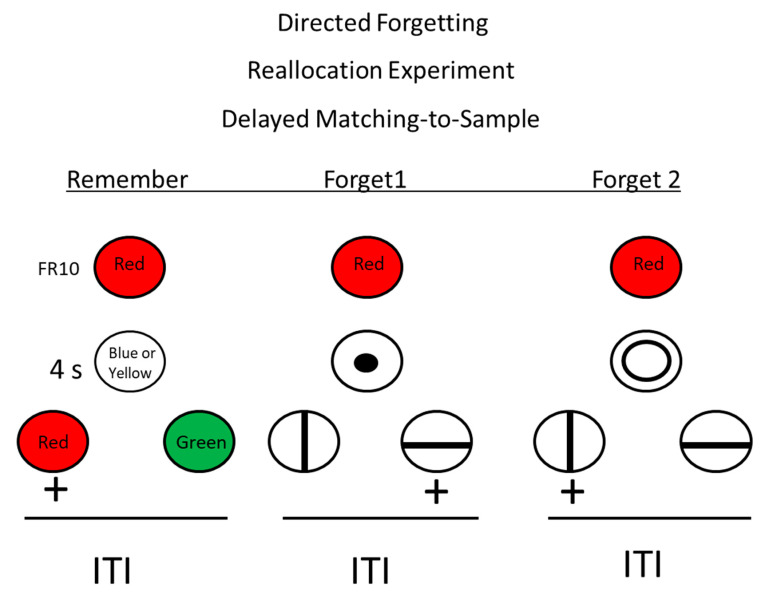
Directed forgetting: reallocation procedure. Pigeons were trained on a red/green delayed matching task. + = reinforcement. During the delay, insertion of an r-cue (remember cue) indicated that they would be tested for their memory for the sample. Insertion of one of two f-cues (forget cues) indicated that they would not be tested for their memory for the sample, but they would have to remember the forget cue (after Roper, Kaiser, and Zentall, 1995) [29].

**Figure 9 animals-13-01165-f009:**
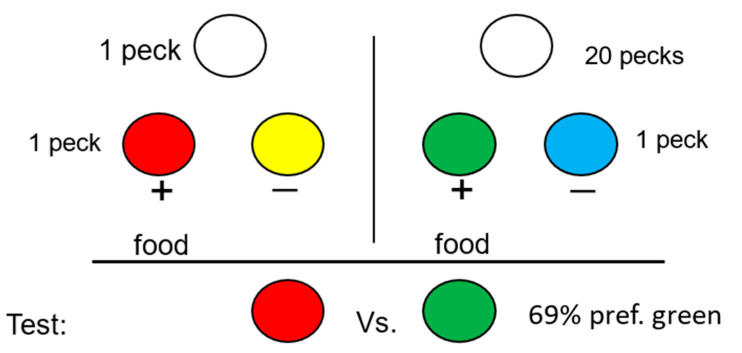
Design of the justification-of-effort experiment with pigeons: On some trials, a single peck was required to present a simple simultaneous (red/yellow) discrimination. On other trials, 20 pecks were required to present a simple simultaneous (green/blue) discrimination. On probe trials, pigeons were given a choice between the two former positive stimuli (after [53]).

**Figure 10 animals-13-01165-f010:**
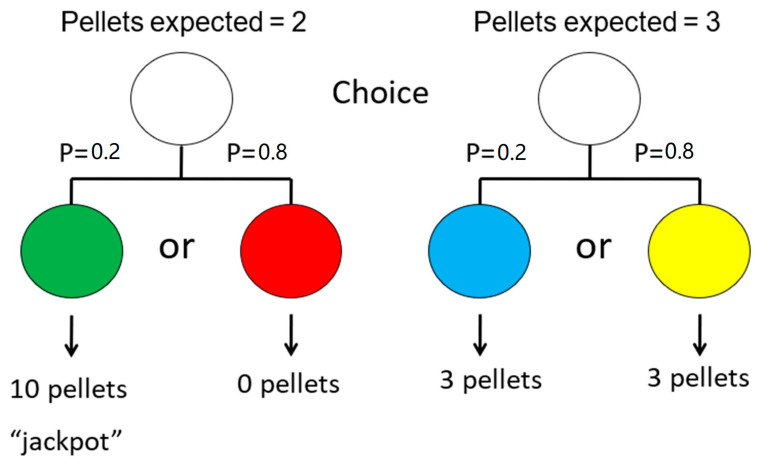
Design of the suboptimal choice experiment with pigeons using differential magnitude of reinforcement. Choice of the left side resulted in a 20% chance of getting a green stimulus, signaling 10 pellets of food, but an 80% chance of getting a red stimulus, signaling no food. Choice of the right side resulted in a 20% chance of getting a blue stimulus or an 80% chance of getting a yellow stimulus, each signaling three pellets of food (after [60]).

## Data Availability

Data and code are available from the author.

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
