# Peer review of "Comparative Cognition Research Demonstrates the Similarity between Humans and Other Animals"

_animals, 2023, doi:10.3390/ani13071165_

Round 1

Reviewer 1 Report

Revision of the paper: Comparative Cognition Research Demonstrates the Similarity Between Humans and Other Animals

Author: T. Zentall

Journal: Animals

MS ID: animals-2178054

This review paper argues that non-human animals, when appropriately tested, show many cognitive abilities that were once considered unique to humans. Likewise, the review suggests simpler mechanisms could be responsible for human behaviours typically attributed to complex cognitive processes. This view is supported by an analysis of the literature on the concepts of sameness and stimulus equivalence, as well as on abilities such as directed forgetting and transitive inference. Moreover, phenomena such as cognitive dissonance, sunken costs effects and unskilled gambling-like behaviours are analysed.

The proposition put forward by this paper, although not entirely new to this field, is fundamental for the advancement of comparative cognition studies. I thus think that this review, which stresses this point with a focus on avian literature, will be of interest to many scholars and could represent an important contribution to the existing literature.

However, I think the Author should try to clarify already in the Abstract why he focused his attention on this specific and apparently miscellaneous collection of phenomena. Also, the choice of focusing mostly on pigeons as an animal model could be justified, at least in the introduction.

Besides that, I have only some minor observations about this Review, which was written by an author whose personal contribution to the field and whose background knowledge is undeniable.

Minor comments:

When the Author describes the parallelism between developmental studies in preverbal infants and animal research, I think it would be appropriate to describe the theory of “Core knowledge systems” developed by the groups of E. Spelke and S. Carey (and the works that have been done to provide empirical support to this theory in another avian species, the domestic chicken).

Please note that transitive inference has been reported also in young domestic chicks.

Author Response

I have included my response to the reviewer under each comment, signaled by an *

This review paper argues that non-human animals, when appropriately tested, show many cognitive abilities that were once considered unique to humans. Likewise, the review suggests simpler mechanisms could be responsible for human behaviours typically attributed to complex cognitive processes. This view is supported by an analysis of the literature on the concepts of sameness and stimulus equivalence, as well as on abilities such as directed forgetting and transitive inference. Moreover, phenomena such as cognitive dissonance, sunken costs effects and unskilled gambling-like behaviours are analysed.

The proposition put forward by this paper, although not entirely new to this field, is fundamental for the advancement of comparative cognition studies. I thus think that this review, which stresses this point with a focus on avian literature, will be of interest to many scholars and could represent an important contribution to the existing literature.

However, I think the Author should try to clarify already in the Abstract why he focused his attention on this specific and apparently miscellaneous collection of phenomena. Also, the choice of focusing mostly on pigeons as an animal model could be justified, at least in the introduction.

*I have made the sample of cognitive phenomena quite diverse to demonstrate their generality and have now indicated that (lines 47-49).

Besides that, I have only some minor observations about this Review, which was written by an author whose personal contribution to the field and whose background knowledge is undeniable.

Minor comments:

When the Author describes the parallelism between developmental studies in preverbal infants and animal research, I think it would be appropriate to describe the theory of “Core knowledge systems” developed by the groups of E. Spelke and S. Carey (and the works that have been done to provide empirical support to this theory in another avian species, the domestic chicken).

*The theory of “core knowledge systems” goes well beyond the relatively simple cognitive behavior that I have outlined in my review. Certainly, children have cognitive abilities that surpass those of pigeons, not the least is language. The current review merely touches on several cognitive abilities that are thought to be uniquely human.

Please note that transitive inference has been reported also in young domestic chicks.

*I now mention that transitive inference has also been found in baby chicks.

Reviewer 2 Report

An interesting and scientifically sound review with the emphasis on “natural concepts” in pigeons, different interpretation of results by alternative research paradigms and possible implications for similar studies on other species.

Therefore, only a couple of comments/considerations.

Similar behaviors may signal sometimes similar processes (lines 37-42 and elsewhere), sometimes different. Even in humans. Not to say about so distant groups as birds/reptiles and humans/mammals. On the other hand, if viewed as a system, an organism may demonstrate different levels of functioning in seemingly similar circumstances, including the lowest possible level. However, I doubt if straightforward interspecies comparisons are possible. Solving such puzzles would necessarily involve evolution, ontogenetic history, social structures and, where appropriate, culture. Also, the development and the structure of a specific mind as a whole is not a minor issue once the problem of comparison is raised. Hence, a cautious wording of possible implications may be more appropriate.

The manuscript is somewhat overloaded with pictures. All reviewed examples of studies had been published, so why not use references for some?

One particular comment on Figure 5: the upper and lower pictures seem to be the same. Is there any mistake? Why not use Figure 1 from the article “Sameness May Be a Natural Concept…”, 2018?

Minors:

Line 158:  use “same/different” or “same-different” instead of “same different”

Line 283: use article “a” instead of “an”

Author Response

Response to reviewr follows each comment and is signaled by an *

An interesting and scientifically sound review with the emphasis on “natural concepts” in pigeons, different interpretation of results by alternative research paradigms and possible implications for similar studies on other species.

Therefore, only a couple of comments/considerations.

Similar behaviors may signal sometimes similar processes (lines 37-42 and elsewhere), sometimes different. Even in humans. Not to say about so distant groups as birds/reptiles and humans/mammals. On the other hand, if viewed as a system, an organism may demonstrate different levels of functioning in seemingly similar circumstances, including the lowest possible level. However, I doubt if straightforward interspecies comparisons are possible. Solving such puzzles would necessarily involve evolution, ontogenetic history, social structures and, where appropriate, culture. Also, the development and the structure of a specific mind as a whole is not a minor issue once the problem of comparison is raised. Hence, a cautious wording of possible implications may be more appropriate.

*I have moderated lines 35-39 to reflect similarities and differences from humans and the somewhat arbitrary selection of research with pigeons.

The manuscript is somewhat overloaded with pictures. All reviewed examples of studies had been published, so why not use references for some?

*I don’t think the reader will bother searching for the original paper

One particular comment on Figure 5: the upper and lower pictures seem to be the same. Is there any mistake? Why not use Figure 1 from the article “Sameness May Be a Natural Concept…”, 2018?

*I think the reviewer is referring to Figure 3. Top refers to matching, bottom refers to mismatching (now noted also in figure caption).

Minors:

Line 158:  use “same/different” or “same-different” instead of “same different”

*Fixed

Line 283: use article “a” instead of “an”